# Exploring the Link Between Mucin 2 and Weaning Stress-Related Diarrhoea in Piglets

**DOI:** 10.3390/ijms26020599

**Published:** 2025-01-12

**Authors:** Li Wang, Long Jin, Liulian Zhang, Xuankai Huang, Ziyu Li, Zhimin Li, Ke Li, Yuan Xu, Shengwei Di, Shiquan Cui, Xibiao Wang

**Affiliations:** 1College of Animal Science and Technology, Northeast Agricultural University, No. 600 Changjiang Road, Xiangfang District, Harbin 150030, China; wanglisy1982@126.com (L.W.); s220501016@neau.edu.cn (L.J.); zhangliulianlh@163.com (L.Z.); ziyuli426@163.com (Z.L.); m18647960742@163.com (Z.L.); abc540409376@126.com (K.L.); xuyuan081086@163.com (Y.X.); dishengwei@neau.edu.cn (S.D.); 2Branch of Animal Husbandry and Veterinary, Heilongjiang Academy of Agricultural Sciences, 2 Heyi St., Longsha District, Qiqihaer 161005, China; hxk87822@163.com

**Keywords:** weaning stress, diarrhoea, *MUC2*, piglets, intestinal injury

## Abstract

To explore the relationship between intestinal mucin 2 (*MUC2*) and weaning-induced diarrhoea in piglets, we analysed *Min* and *Landrace* piglets. The piglets were divided into a healthy weaned group, a weaned diarrhoea group, and a healthy unweaned control group. Intestinal tissues were collected, and goblet cell numbers, sizes, and degrees of intestinal injury were observed and recorded. Intestinal tissue *MUC2* mRNA and protein expression were analysed via quantitative real-time PCR (qRT–PCR) and Western blotting. *Min pigs* presented significantly lower diarrhoea rates and intestinal injury scores than *Landrace pigs* (*p* < 0.01). The intestinal injury scores in the weaned diarrhoea group were significantly greater than those in the unweaned groups (*p* < 0.05), with *Min pigs* consistently exhibiting lower injury scores than *Landrace pigs*. Specifically, unweaned *Min pigs* presented significantly greater duodenal *MUC2* mRNA (*p* < 0.05), and weaned healthy *Min pigs* presented notably greater expression in both the duodenum and jejunum (*p* < 0.01). These findings reveal enhanced intestinal protection against weaning stress and diarrhoea in *Min pigs*, with elevated *MUC2* levels likely contributing to lower injury scores and milder symptoms, thus highlighting the influence of genetic differences.

## 1. Introduction

Weaning is an inevitable stage in growth but can disrupt internal homeostasis due to separation from sows and changes in nutrition and the environment [1,2]. This imbalance can lead to adverse symptoms, particularly weaning-induced diarrhoea [3], which is the primary stress-related disease affecting piglet survival and growth. According to relevant statistics, the incidence of diarrhoea in piglets exceeds 50%, and diarrhoea-related deaths account for 39.8% of total piglet mortality. The imbalance in intestinal physiological homeostasis caused by weaning stress is a major factor leading to intestinal mucosal damage and subsequent weaning-related diarrhoea in piglets [4,5]. Therefore, the developmental status of the intestinal mucosa during weaning directly affects the incidence and severity of diarrhoea during weaning.

Mucins, as key components of the mucus layer, are a family of high-molecular-weight, heavily glycosylated proteins that are widely distributed across various tissues and organs in the body [6]. More than 20 types of mucins have been identified, categorised as secreted or membrane-bound [7]. Secreted mucins, including *MUC2*, *MUC5AC*, *MUC5B*, *MUC6*, and *MUC19*, form a protective mucus gel on intestinal epithelial cells [8,9]. *MUC2*, which is synthesised and secreted by goblet cells, is the most important protein constituting the intestinal mucus layer. It is widely expressed in the intestine, and changes in its expression level and structure are closely associated with the onset and development of intestinal mucosal barrier damage [10,11]. The key region of *MUC2* is a protein core domain composed of a repetitive sequence rich in proline, threonine, and serine (PTS domain) [12]. This domain forms a polysaccharide–protein complex upon binding with polysaccharides, and through O-glycan chains, it combines with water to endow the mucus with gel-like properties [13,14]. The degree of glycosylation of the O-linked oligosaccharide chains in mucins largely determines their protective effects on intestinal epithelial cells [15]. Once released into the intestinal lumen, *MUC2* forms a polymer network, ensuring structural stability and serving as a binding site for secretory immunoglobulin A (sIgA) and antimicrobial peptides, enhancing immune barrier function [16,17]. *MUC2* also supports the transport and absorption of nutrients [18], highlighting its clinical importance in maintaining intestinal health.

*Min pigs* are known for their high reproductive performance, good meat quality, strong disease resistance, and adaptability to rough feeding. This study aimed to investigate whether *Min pigs* possess a more robust intestinal mucus barrier and immune function than other breeds and to determine the role of these intestinal characteristics in their susceptibility to postweaning diarrhoea.

## 2. Results

### 2.1. Diarrhoea in Piglets Within One Week After Weaning

The diarrhoea rate, diarrhoea frequency, and diarrhoea index of Min and *Landrace piglets* within one week after weaning are shown in Table 1. The experimental results indicate that *Min pigs* had a significantly lower diarrhoea rate, diarrhoea frequency, and diarrhoea index than *Landrace pigs* (*p* < 0.0001).

### 2.2. Intestinal Tissue Pathological Injury Scores

The intestinal mucosa is an important barrier against external environmental stimuli and pathogen invasion. As shown in Figure 1, after weaning, the intestinal mucosal structure of piglets changes, characterised by villus atrophy and rupture. As shown in Figure 2, compared with those of the unweaned group, except for slight decreases in the damage scores in the jejuna of the weaned healthy *Landrace* group, the intestinal damage scores of both the *Min* and *Landrace* piglets after weaning were greater, and the diarrhoea groups had significantly (*p* < 0.05) higher intestinal damage scores than the unweaned group. A comparison of the breeds in each group revealed that the intestinal damage scores of the *Landrace pigs* were greater than those of the *Min pigs* across all the intestinal segments. Furthermore, in the weaned diarrhoea group, the duodena and jejuna of the *Landrace pigs* had significantly greater intestinal damage scores than those of the *Min pigs* (*p* < 0.05). No significant differences were detected between breeds in the other groups.

### 2.3. Comparison of the Numbers and Volumes of Goblet Cells in Piglet Intestines

Goblet cells are important secretory cells distributed in the intestines. The mucins they synthesise and secrete are the major components of the intestinal mucus barrier and play crucial roles in defending against external invasions and maintaining intestinal homeostasis. As shown in Figure 3, many goblet cells were distributed in the intestine and clearly outlined, with their cytoplasm filled with red-stained mucus granules after PAS staining.

As shown in Figure 4, compared with those in the unweaned group, the number of goblet cells in the duodena, jejuna, ilea, and colons of the *Landrace pigs* decreased after weaning, and the goblet cell counts in the duodena and ilea of the weaned diarrhoea group were significantly lower than those in the unweaned group (*p* < 0.01). In *Min pigs*, the changes in goblet cell numbers across the intestinal segments were similar, with the weaned healthy group showing greater numbers than the unweaned group, whereas the diarrhoea group presented lower numbers than the unweaned group. Specifically, the *Min pig* goblet cell count in the weaned diarrhoea group was significantly lower (*p* < 0.05) than that in the jejuna and colons of the unweaned group. A comparison of the breeds revealed that the goblet cell counts in the duodena (*p* < 0.01), jejuna (*p* < 0.01), ilea (*p* < 0.05), and colons in the healthy weaned *Min pigs* were significantly greater than those in the *Landrace pigs*.

As shown in Figure 5, after weaning, the volumes of goblet cells in both *Landrace pigs* and *Min pigs* changed in different intestinal segments. Compared with those of the unweaned group, the goblet cell volumes in the duodena, jejuna, and ilea of the weaned healthy group were greater in both the *Landrace* and *Min pigs*. Specifically, the goblet cell volume in *Landrace pigs* increased by 19.57%, 2.75%, and 23.72%, whereas in *Min pigs*, the goblet cell volume increased by 29.26%, 62.83%, and 5.83%, respectively. However, the goblet cell volume in the colons decreased by 19.41% in *Landrace pigs* and 7.00% in *Min pigs* in the healthy weaned group. Compared with those of the unweaned piglets, the goblet cell volumes in the duodena, jejuna, and ilea of both the *Landrace* and *Min pigs* in the weaned diarrhoea group were lower. In *Landrace pigs*, the goblet cell volume decreased by 21.37%, 25.46%, and 1.84%, whereas in *Min pigs*, it decreased by 1.48%, 36.34%, and 13.81%, respectively. However, the goblet cell volume in the colons increased by 23.21% in *Landrace pigs* and by 34.97% in *Min pigs* in the weaned diarrhoea group. In both breeds after weaning, the goblet cell volumes in *Min pigs* were greater than those in *Landrace pigs* across all the intestinal segments, except for the weaned diarrhoea group, in which *Min pigs* had smaller goblet cell volumes in their jejuna. Moreover, the goblet cell volumes in the duodena and jejuna of *Min pigs* in the healthy weaned group were significantly greater (*p* < 0.05) than those in *Landrace pigs*.

### 2.4. Intestinal MUC2 mRNA Expression Levels

As shown in Figure 6, *MUC2* mRNA expression in the jejuna of *Min pigs* in the postweaned healthy group was significantly greater than that in the unweaned group (*p* < 0.05), whereas *MUC2* mRNA expression in the diarrhoea group was significantly lower than that in the postweaned healthy group (*p* < 0.01). Compared with that in the unweaned group, *MUC2* mRNA expression in *Landrace pigs* increased in all postweaning groups, but the increase was not statistically significant. *MUC2* mRNA expression in the ilea and colons of both the *Landrace* and *Min pigs* in the postweaning healthy and diarrhoea groups decreased compared with that in the unweaned group, and *MUC2* mRNA expression in the colons of the postweaning diarrhoea groups was significantly lower than that in the unweaned group (*p* < 0.05). Compared with *Landrace pigs*, *Min pigs* presented significantly greater *MUC2* mRNA expression across all intestinal segments in the unweaned group, with duodenal *MUC2* mRNA expression in *Min pigs* being significantly greater than that in the duodena of *Landrace pigs* (*p* < 0.01). In the postweaned healthy group, *Min pigs* presented higher duodenal, jejunal, and colonic *MUC2* mRNA expression than *Landrace pigs* did, with the duodena and jejuna showing significant differences (*p* < 0.01).

### 2.5. Intestinal MUC2 Protein Expression Levels

As shown in Figure 7B, duodenal *MUC2* protein expression in the healthy weaned *Min pig* group was slightly increased compared with that in the unweaned group, whereas in the healthy *Landrace* weaned and diarrhoeal groups, *MUC2* protein expression was decreased. Jejunal *MUC2* protein expression in the unweaned *Min pig* group was significantly greater than that in the healthy weaned *Min pig* group and diarrhoea group (*p* < 0.05). There was no significant difference in *MUC2* expression between the healthy *Landrace* group and the unweaned group, but the diarrhoea group presented significantly greater *MUC2* expression than the unweaned group (*p* < 0.05) or the healthy weaned group (*p* < 0.01). No significant changes in ileal *MUC2* protein expression were observed in either breed. Colonic *MUC2* protein expression was lower in the healthy weaned groups of each breed than in the unweaned groups, whereas *MUC2* protein expression was increased in the diarrhoea groups.

Among the breeds, *MUC2* protein expression in the healthy unweaned and weaned *Min pigs* was greater than that in the *Landrace pigs*, with a significant difference in the duodena (*p* < 0.01). Across the weaned diarrhoea groups, jejunal and duodenal *MUC2* protein expression in the *Min pigs* was lower than that in the *Landrace pigs*, whereas the *Min pigs* presented higher *MUC2* protein expression in the other intestinal segments, with a significant difference in the colon (*p* < 0.01).

### 2.6. Intestinal MUC2 Expression Localisation

Piglet intestinal *MUC2* immunohistochemistry results are shown in Figure 8. The intestinal *MUC2* protein expression levels changed from before to after weaning, but there was no significant change in *MUC2* protein localisation due to weaning stress. Microscopic observation revealed that the *MUC2* protein expression at the duodenal, jejunal, and ileal sites was generally consistent and was primarily concentrated in the epithelial layer of the small intestinal villi. Additionally, there was diffuse *MUC2* protein expression in the mucosal lamina propria, and a small amount of *MUC2* protein was localised in the small intestinal glands. The *MUC2* protein expression level gradually decreased from the intestinal mucosal epithelium layer inwards.

## 3. Discussion

Pig breed is an important factor influencing the response to weaning stress [19,20]. In this study, *Min pigs* had significantly lower diarrhoea incidence, frequency, and score than *Landrace pigs* did, indicating that *Min pigs* have better resistance to weaning-stress-induced diarrhoea than *Landrace pigs*. Histological analysis revealed varying degrees of intestinal damage in both breeds after weaning, with the small intestine being the most affected. Damage included villous rupture, epithelial necrosis, and crypt hyperplasia, with diarrhoea groups exhibiting more severe damage. *Min pigs* consistently presented lower intestinal damage scores than *Landrace pigs*. The integrity of the intestinal mucosa is crucial for nutrient absorption and defence against harmful substances [21,22]. Studies have shown that weaning stress can cause acute or chronic changes in intestinal structure and function, leading to villous atrophy, rupture, and increased crypt depth [23,24,25]. Weaning stress leads to a reduction in villus height and an increase in crypt depth [26]. Overall, weaning stress damages the intestinal mucosa in piglets, causing unfavourable changes in intestinal morphology [27]. This disruption impairs nutrient and water absorption, contributing to diarrhoea in piglets [28].

The mucus layer formed by mucins, primarily *MUC2*, is secreted by goblet cells and serves as the first line of intestinal defence by preventing pathogen invasion and maintaining mucosal integrity [29,30,31]. Damage to goblet cells reduces mucin secretion, weakening the mucus barrier and increasing susceptibility to inflammation [32,33]. Studies have linked decreased mucin synthesis to goblet cell depletion in conditions such as inflammatory bowel disease and necrotising enterocolitis [34,35,36]. Goblet cells also play an essential role in neonatal intestinal immunity before passive immunity is established [37].

After weaning, the numbers and volumes of duodenal, jejunal, and ileal goblet cells were reduced in both the *Min pig* and *Landrace* diarrhoea groups. In the healthy *Min pig* group, the number of goblet cells was greater than that in the diarrhoea group, whereas in the *Landrace piglets*, the number of goblet cells was significantly lower, increasing the risk of intestinal damage. Although the number of goblet cells decreased in the colons of the diarrhoea group, their volume increased, possibly as a compensatory mechanism to offset mucus loss [38]. However, this compensatory increase in secretion may overload the endoplasmic reticulum, causing endoplasmic reticulum stress and ultimately leading to goblet cell apoptosis [39,40], further reducing the number of goblet cells. Compared with *Landrace pigs*, healthy weaned *Min pigs* presented greater numbers and volumes of goblet cells, especially in the duodenum and jejunum. These findings suggest that *Min pigs* can maintain a higher level of mucin secretion after weaning, providing more effective protection against intestinal damage caused by weaning stress.

*MUC2* is the most important glycoprotein in the intestinal mucus layer and serves as a marker of intestinal epithelial permeability [41,42]. The main protein structure of *MUC2* not only lubricates and protects the intestinal mucosa but also provides binding sites on its sugar chains for the normal gut microbiota and immunoglobulins [16,43]. It is also involved in intercellular signalling and the regulation of intestinal mucosa-associated immune factors [17,44,45]. Studies have shown that the loss of *MUC2* expression in the intestinal mucosa can lead to intestinal diseases such as bacterial diarrhoea [46,47]. Research by Rasmussen et al. [48] and Puiman et al. [49] indicated that decreased *MUC2* expression in the intestines of preterm piglets weakens the mucosal barrier and increases susceptibility to NEC. Additionally, Juxing Chen et al. [50] reported reduced *MUC2* expression in the intestines of broilers with impaired gut barrier function. *MUC2* not only is vital for maintaining mucosal integrity in the context of weaning-induced diarrhoea but also has been associated with other stress-related conditions, such as necrotising enterocolitis and heat stress [51,52,53,54,55]. In these cases, *MUC2* plays a crucial role in maintaining mucosal integrity and preventing systemic inflammation [16]. These findings highlight *MUC2* as a universal indicator of intestinal barrier function [56], with potential applications for enhancing gut health not only in various livestock species but also in humans [57].

In this study, we examined the changes in *MUC2* mRNA and protein expression in the intestines of piglets before and after weaning. Jejunal patterns of expression varied between *Min pigs* and *Landrace pigs*. Compared with those in the unweaned and weaned healthy groups, the *MUC2* mRNA and protein levels in the *Min pig* diarrhoea group were lower, whereas those in the *Landrace* diarrhoea group were greater than those in both healthy groups. Since the jejunum is the longest segment of the small intestine, these divergent changes in jejunal *MUC2* expression between *Min pigs* and *Landrace pigs* may be more closely related to weaning diarrhoea [3]. The inflammatory response triggered by weaning stress may be more pronounced in *Landrace pigs*. Although the *Landrace* diarrhoea group presented greater goblet cell numbers and volumes than the *Min pig* group did, acute *MUC2* secretion did not significantly improve their defence ability, as *Min pigs* presented lower diarrhoea rates and indices. Healthy *Min pigs* presented greater goblet cell numbers, volumes, and *MUC2* expression, whereas diarrhoeal *Min pigs* presented lower levels than *Landrace pigs* did. These findings suggest that complex genetic and stress response mechanisms may be involved in mucin secretion regulation, and future studies could investigate these mechanisms to reveal the differences in intestinal barrier function between the two breeds. Overall, this study provides insights into how different pig breeds respond to weaning stress and offers potential applications for the genetic improvement of pigs. By identifying genetic markers associated with *MUC2* expression, pigs with stronger intestinal immune function could be bred to better cope with weaning stress and intestinal diseases. However, this study focused solely on weaning stress as a model and did not include other stress factors, such as malnutrition or pathogenic microbial infections. Therefore, future research should consider more complex stress models to better reflect the combined impact of multiple stressors on gut health in real-world production environments. Additionally, integrating *MUC2* regulation into nutritional strategies such as the inclusion of prebiotics or amino acids to support mucus production could help mitigate stress-induced intestinal damage [58]. Extending the experimental period could provide further insights into the recovery process of intestinal barrier function and its timing, offering a theoretical foundation for the development of antistress strategies and improving intestinal health in pigs. Investigating the role of *MUC2* in these complex stress models will reveal its broader applicability in promoting livestock health and productivity.

## 4. Materials and Methods

### 4.1. Animal Feeding and Husbandry

In this study, we selected *Min pigs* and *Landrace piglets* from the same batch born at a breeding farm in Lanxi County, Heilongjiang Province. Both Min and *Landrace* sows and their piglets were raised under the same environmental and immunisation conditions. Before rearing, the farrowing pens were thoroughly cleaned and disinfected. The piglets were housed with the sow in the farrowing pens until weaning at 35 days of age. To minimise environmental stress, the piglets were observed until 38 days of age and were reared in the farrowing pens after weaning. The stocking density was maintained at approximately 0.5 m^2^ per piglet. The pens were equipped with heating lamps to ensure a temperature of 28–30 °C, and the relative humidity was controlled at 60–70%. The piglets were fed a standard commercial diet twice daily, and clean drinking water was provided ad libitum. The dietary nutrient levels were as follows: crude protein, 17.5%; net energy, 1.4 MJ/kg; lysine, 1.4%; total calcium, 0.6%; and phytate phosphate, 0.35%. Feed and water cleanliness were ensured throughout the experimental period.

A total of 108 piglets (48 *Min piglets* and 60 *Landrace piglets*) were randomly selected for the study. The farrowing pens and feeding equipment were cleaned and disinfected regularly to prevent disease. No antibiotics or probiotics were administered to avoid influencing the gut microbiota. On the basis of the diarrhoea scores recorded in the first 3 days after weaning, piglets with no diarrhoea or other symptoms were classified into the healthy group, whereas those with diarrhoea records were classified into the diarrhoea group. Healthy unweaned piglets were used as the control group. In the weaned groups (healthy and diarrhoea), 6 piglets were randomly selected and slaughtered (3 *Min pigs* and 3 *Landrace pigs*), and 3 healthy unweaned piglets from each breed were also slaughtered as controls. A total of 18 piglets were slaughtered, as shown in Table 2.

### 4.2. Sample Collection

Following the experimental methods of Wen et al. [59], each piglet’s faeces and anus were observed daily at 9:00 a.m. and 4:00 p.m. Faeces were scored on the basis of their appearance (firm or loose) and the presence of redness or faecal contamination around the anus (scoring criteria are shown in Table 3). The diarrhoea rate, diarrhoea frequency, and diarrhoea index were calculated to reflect the degree of diarrhoea in the piglets.Diarrhoea rate = (number of piglets with diarrhoea/total number of piglets) × 100%(1)Diarrhoea frequency = Σ (number of piglets with diarrhoea × diarrhoea days per piglet)/(total number of test piglets × duration of the trial) × 100%(2)Diarrhoea index = sum of faecal scores/total number of test piglets(3)

After the piglets were slaughtered, approximately 0.5 g tissue samples were collected from the duodenum, jejunum, ileum, and colon. These samples were rinsed with cold physiological saline to remove blood and faeces. One sample was placed into an EP tube treated with DEPC, frozen in liquid nitrogen, and stored at −80 °C for later use. The other sample was dried with filter paper, placed into a sealed bag, and stored at −20 °C. Additionally, 1 cm × 1 cm tissue samples were collected, rinsed with cold physiological saline to remove blood and faeces, and preserved in 10% formaldehyde to generate paraffin-embedded tissue sections.

### 4.3. Histological Examination

#### 4.3.1. Preparation of Paraffin Sections

After the fresh tissue was fixed in fixative for 24 h, dehydration was performed sequentially using graded ethanol (75%, 85%, 90%, 95%) and absolute ethanol. The tissue was then cleared with xylene and treated with an alcohol–xylene mixture. Afterwards, the tissue was immersed in melted paraffin at 65 °C for 1 h. The tissue was embedded in an embedding machine (JB-P5, Wuhan Junjie Electronic Co., Wuhan, China) and cooled. The tissue was subsequently sectioned into 4 μm thick slices via a paraffin microtome (RM2016, Leica Instruments, Shanghai, China). The sections were flattened in 40 °C warm water, bound to slides, and dried in a 60 °C oven. Finally, the sections were stored at room temperature for further use.

#### 4.3.2. Haematoxylin and Eosin (H&E) Staining

The duodenal, jejunal, ileal, and colonic tissue sections were sequentially dehydrated and cleared with xylene, absolute ethanol, and 75% alcohol. After being stained with haematoxylin (G1003 Servicebio, Wuhan, China), differentiated, and blued, they were dehydrated again with gradient alcohol, stained with eosin (G1003 Servicebio, China), dehydrated with ethanol, and cleared with xylene before being mounted with neutral resin. The sections were then observed and photographed under a Nikon DS-Ri1 microscope at 100× magnification for the duodenum, jejunum, and ileum and at 40× magnification for the colon.

Three sections from the duodenum, jejunum, ileum, and colon were collected from each piglet. For each section, six different fields of view were randomly selected. Damage, such as villus shedding, rupture, and atrophy in the duodenum, jejunum, and ileum, was scored according to the methods of Chiu et al. [60]. The degree of mucosal damage and crypt injury in the colon was scored on the basis of the methods of Dieleman et al. [61].

#### 4.3.3. Periodic Acid–Schiff (PAS) Staining

The process of dehydration for the duodenum, jejunum, ileum, and colon tissue sections was the same as that described previously. After being stained with PAS-B solution (G1008, Servicebio, China), the sections were washed with water. Then, they were differentiated with PAS-A solution, counterstained with haematoxylin, and washed with water again. Finally, the sections were dehydrated, cleared, and mounted with neutral balsam. The sections were observed and photographed under a Nikon DS-Ri1 microscope (×100).

For the quantification of goblet cells, the method described by Desantis et al. [62] was used. Three sections were taken from each segment of the small intestine of each piglet. Goblet cells were randomly selected from six different fields via ImageJ software, version 1.54g on the basis of colour intensity. To measure the cell diameters, the method described by Sjostrom et al. [63] was used. Three sections of intestinal tissue were obtained from each piglet, and in each section, six fields were randomly selected. In each of these fields, 80 goblet cells were randomly chosen, and their long and short axes were measured via ImageJ software. The geometric means of the long and short axes represent the diameters of the goblet cells in that tissue. The volume of each cell was calculated via the following formula:Average volume: V = π/6∑fiDi3/∑fi(4)
where Di is the cell diameter and fi represents the number of goblet cells with a diameter of Di.

#### 4.3.4. Immunohistochemistry (IHC)

Paraffin-embedded tissue sections of the duodenum, jejunum, ileum, and colon were deparaffinized and rehydrated following standard procedures. Antigen retrieval was performed with a sodium citrate solution (G1202, Servicebio, China), followed by blocking with a peroxidase blocker. The sections were then blocked with 3% BSA (G5001, Servicebio, China). A primary antibody against *MUC2* (1:100; A14659, ABclonal, Wuhan, China) was incubated with the samples overnight at 4 °C. The secondary antibody, HRP-conjugated goat anti-rabbit IgG (1:2000; AS014, ABclonal, Wuhan, China), was incubated with the samples at room temperature for 50 min. DAB (G1211, Servicebio, China) was used for colour development, and haematoxylin was added for 3 min for counterstaining. The sections were then dehydrated, cleared, and mounted with neutral resin. Observations and imaging were performed under a Nikon DS-Ri1 microscope at 100× magnification.

### 4.4. qRT–PCR

The duodenum, jejunum, ileum, and colon tissues were minced, and total mRNA from each intestinal sample was extracted via TRIzol reagent (Invitrogen, Beijing, China). The concentrations and purities of the nucleic acids were determined via a UV spectrophotometer (Μltrospec 1000, Pharmacia, New Jersey, NJ, USA). Total mRNA was reverse transcribed into cDNA according to the instructions of the reverse transcription kit (RR047A, TaKaRa, Beijing, China). The qRT–PCRs were performed via QuantStudio TM software, v 1.5.1 (Applied Biosystems, Waltham, MA, USA) to determine gene expression. The reaction system (10 μL) consisted of 5 μL of FastStart Universal SYBR Master Mix (ROX) (46660900, Roche, Basel, Switzerland), 0.3 μL each of forwards and reverse primers, 3.4 μL of RNase/DNase-free water, and 1 μL of cDNA template. The PCR cycling program included 95 °C for 10 min of initial denaturation, followed by 40 cycles of 95 °C for 15 s (denaturation), 60 °C for 1 min (annealing), and 72 °C for 30 s (extension). Each sample was tested three times. The relative gene expression levels were calculated via the 2^−ΔΔCt^ method.

The primers used in this study were designed and synthesised by Shanghai Shenggong Biological Engineering Co., Ltd. (Shanghai, China) β-actin was used as the reference gene. The primer sequences are shown in Table 4.

### 4.5. Western Blot

Duodenal, jejunal, ileal, and colonic tissues were minced and lysed on ice for 10 min in IP lysis buffer (P0013, Beyotime, Haimen, China) for Western blot analysis. The lysates were subsequently centrifuged at 12,000× *g* for 15 min at 4 °C, after which the supernatants were collected. Protein concentrations were determined via a BCA protein assay kit (P0010S, Beyotime). Protein samples were separated via 7.5–10% SDS–PAGE and transferred onto a polyvinylidene fluoride (PVDF) membrane activated with anhydrous methanol. The membrane was blocked with 5% skim milk at 37 °C for 2 h, incubated with a diluted primary antibody at 4 °C overnight, and then incubated with an HRP-conjugated goat anti-rabbit secondary antibody at room temperature for 1 h. The protein bands were visualised via a hypersensitive ECL chemiluminescence kit (P0018S, Beyotime) and observed via a gel imaging system (NIH, Bethesda, Rockville, MD, USA).

The antibodies used in this study included those against pig β-actin (1:1000; AC026, ABclonal, Wuhan, China), *MUC2* (1:1000; A14659, ABclonal, Wuhan, China), and HRP goat anti-rabbit IgG (1:2000; AS014, ABclonal, Wuhan, China).

### 4.6. Data Processing

All the data were initially processed and calculated via Microsoft Excel. Differences in the goblet cell counts, volumes, intestinal mucosal injury scores, and *MUC2* mRNA and protein expression among different breeds and groups of piglets were analysed via SPSS 25 statistical software with analysis of variance (ANOVA), and Duncan’s multiple range test was employed for post hoc comparisons. The results are expressed as the means ± standard deviations (means ± SDs). GraphPad Prism 8.0 was used to create figures and charts, and significant differences are denoted as * for *p* < 0.05, ** for *p* < 0.01, and *** for *p* < 0.001.

## 5. Conclusions

Compared with *Landrace pigs*, *Min pigs* have a greater ability to resist weaning-stress-induced diarrhoea. The number and volume of goblet cells in the intestines of *Min pigs* were greater than those in the intestines of *Landrace pigs*, and the *MUC2* protein expression levels in the intestines of *Min piglets* were greater than those in the intestines of *Landrace piglets*. These factors are important physiological bases for the lower degree of intestinal damage and the reduced severity of diarrhoea observed in *Min piglets* after weaning.

## Figures and Tables

**Figure 1 ijms-26-00599-f001:**
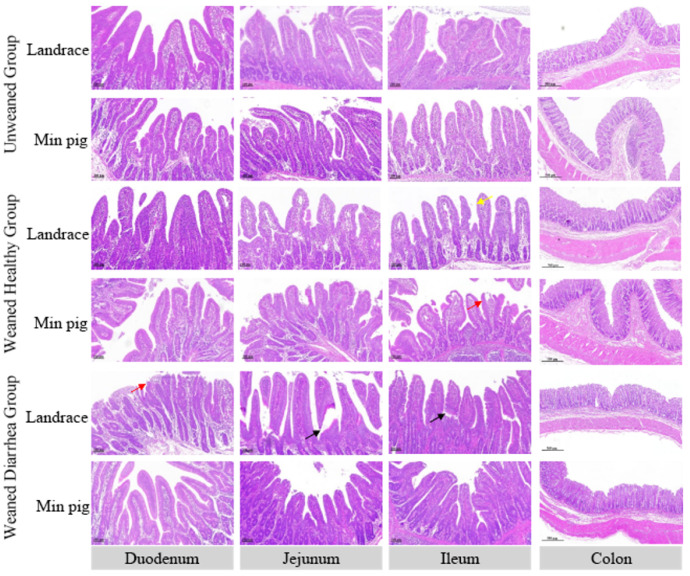
Intestinal structures of *Landrace* and *Min pigs*. In the weaning diarrhoea group, *Landrace pigs* presented significant villi rupture (black arrows) and mucosal shedding (red arrow). Healthy weaned *Landrace pigs* exhibit mucosal shedding (red arrow), whereas *Min pigs* exhibit less damage and villi atrophy (yellow arrow). The images were taken under a microscope with a magnification of 100×.

**Figure 2 ijms-26-00599-f002:**
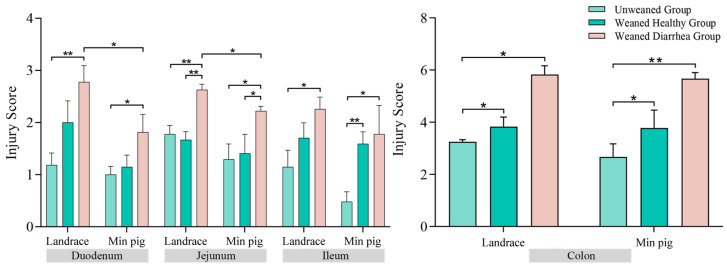
Injury scores of the duodenum, jejunum, ileum, and colons in *Landrace* and *Min pigs*. After weaning, both *Min* and *Landrace* piglets presented increased intestinal damage scores, with the diarrhoea groups showing significantly higher scores than the unweaned group, * *p* < 0.05, ** *p* < 0.01.

**Figure 3 ijms-26-00599-f003:**
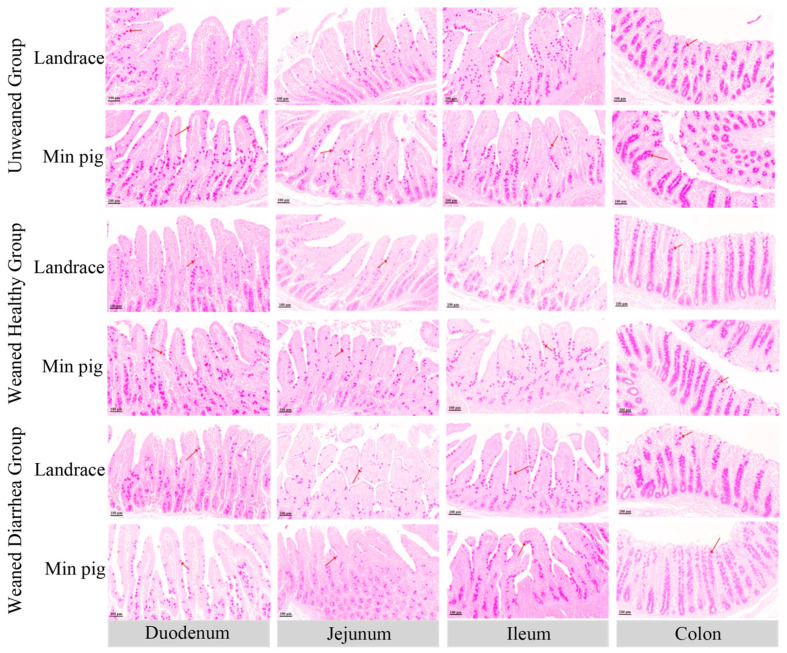
Intestinal goblet cells of *Landrace* and *Min pigs*. PAS staining revealed numerous well-defined goblet cells in the intestine, with their cytoplasm filled with red-stained mucus granules. The red arrows indicate PAS-stained goblet cells. The images were taken under a microscope with a magnification of 100×.

**Figure 4 ijms-26-00599-f004:**
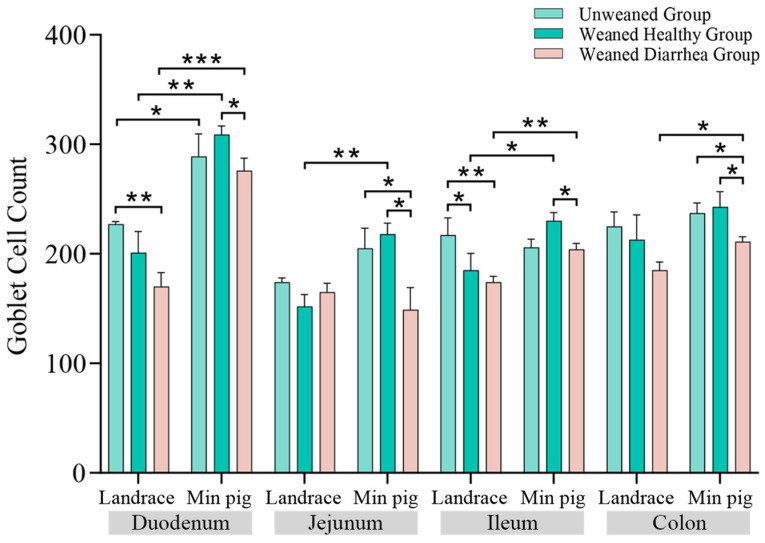
Numbers of goblet cells in the intestines of *Landrace* and *Min pigs*. After weaning, *Landrace pigs* presented a decrease in the number of goblet cells, whereas *Min pigs* presented higher goblet cell counts, indicating better intestinal health, * *p* < 0.05, ** *p* < 0.01, *** *p* < 0.001.

**Figure 5 ijms-26-00599-f005:**
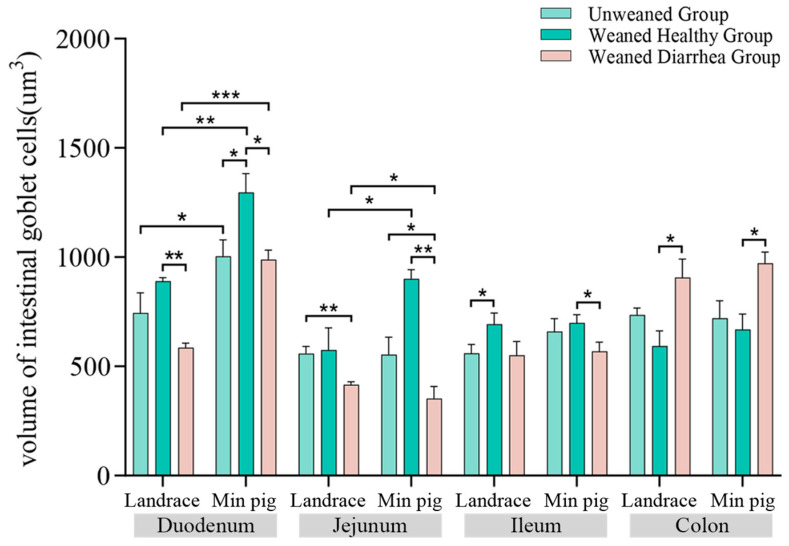
Volumes of goblet cells in the intestines of *Landrace* and *Min pigs*. Volumes of goblet cells in the intestines of *Landrace* and *Min pigs*. After weaning, the goblet cell volume increased in healthy pigs but decreased in the diarrhoea group. *Min pigs* presented greater goblet cell volumes than *Landrace pigs* did, especially in the duodenum and jejunum, * *p* < 0.05, ** *p* < 0.01, *** *p* < 0.001.

**Figure 6 ijms-26-00599-f006:**
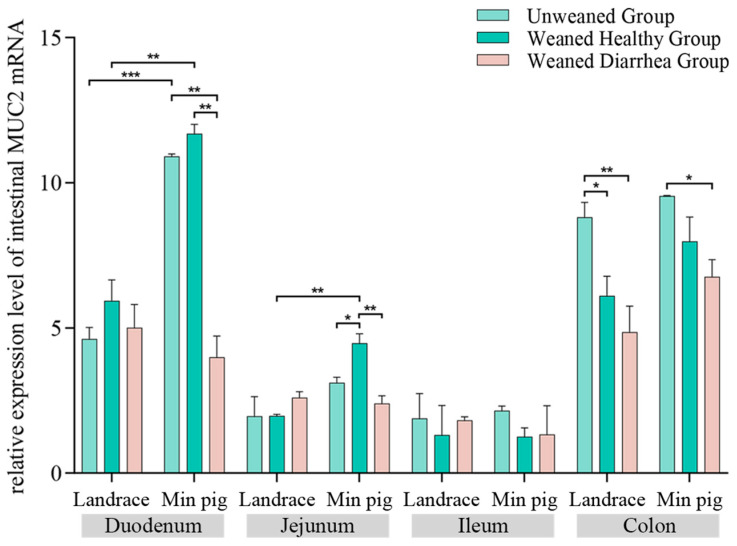
Intestinal *MUC2* mRNA expression pattern. *Min pigs* presented increased *MUC2* expression in the jejunum after weaning, whereas *MUC2* expression decreased in the diarrhoea group. *Min pigs* presented greater overall *MUC2* expression than *Landrace pigs* in all segments, * *p* < 0.05, ** *p* < 0.01, *** *p* < 0.001.

**Figure 7 ijms-26-00599-f007:**
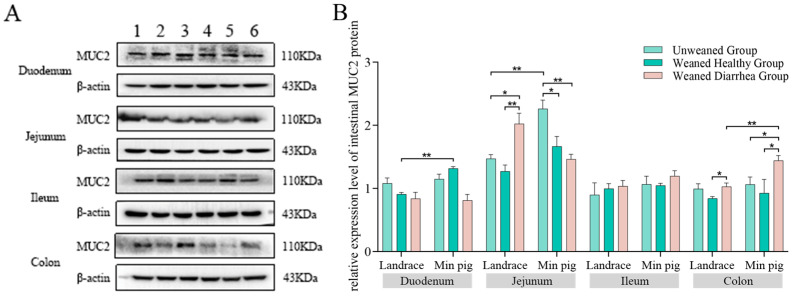
Protein bands and relative *MUC2* expression levels in the duodena, jejuna, ilea, and colons: (**A**) Protein bands. (**B**) Relative expression levels. *Min pigs* presented increased duodenal *MUC2* expression after weaning, whereas *Landrace pigs* presented decreased *MUC2* expression. Overall, *Min pigs* presented greater *MUC2* expression than *Landrace pigs* did, especially in the duodenum. * *p* < 0.05, ** *p* < 0.01.

**Figure 8 ijms-26-00599-f008:**
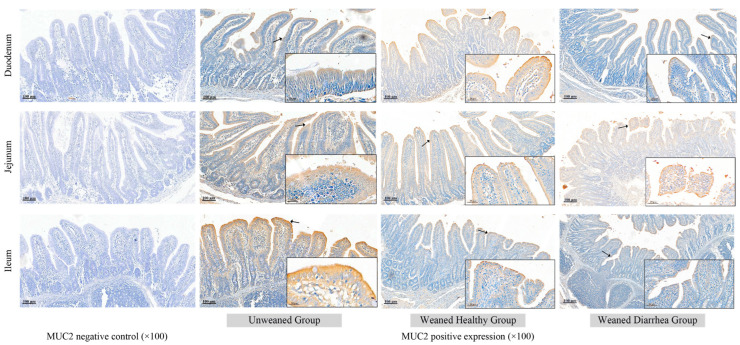
Intestinal *MUC2* protein expression and localisation in the unweaned group. *MUC2* was expressed primarily in the epithelial layer of the villi and lamina propria of the small intestine. No significant localisation changes occurred due to weaning stress. The black arrows indicate the regions of MUC2 protein expression.

**Table 1 ijms-26-00599-t001:** Diarrhoea conditions of *Min* and *Landrace* piglets within one week after weaning.

	Diarrhoea Rate (%)	Diarrhoea Frequency (%)	Diarrhoea Index (%)
*Min pig*	35.42	6.51	0.96
*Landrace pig*	73.33	23.13	2.80
χ^2^	27.80	44.40	
	*p* < 0.0001	*p* < 0.0001	

**Table 2 ijms-26-00599-t002:** Experimental groups.

Weaned Diarrhoea Group		Weaned Healthy Group		Unweaned Group	
*Min pig*	*Landrace*	*Min pig*	*Landrace*	*Min pig*	*Landrace*
3	3	3	3	3	3

**Table 3 ijms-26-00599-t003:** Scoring criteria for diarrhoea conditions.

Diarrhoea Severity	Faecal Appearance	**Score**
Normal	Formed or pellet-like	0
Mild	Soft but formable	1
Moderate	Viscous, unformed, with no separation of faecal water	2
Severe	Liquid, unformed, with separation of faecal water, mucus stools, or pus stools	3

**Table 4 ijms-26-00599-t004:** Primer sequences.

Gene	Primer Sequences (5′-3′)	Product Length	GenBank Accession
*MUC2*	F: CACCACCACCAGCACCACTTG R: TCGGACCAGACGCAGCAGAG	84 bp	XM_021082584.1
β-actin	F: ATGCTTCTAGGCGGACTGT R: CCATCCAACCG ACTGCT	211 bp	AY550069

## Data Availability

The data presented in this study are available upon request from the corresponding author. The availability of the data is restricted to investigators based in academic institutions.

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
