# Peer review of "Exploring the Link Between Mucin 2 and Weaning Stress-Related Diarrhoea in Piglets"

_ijms, 2025, doi:10.3390/ijms26020599_

Round 1
Reviewer 1 Report
Comments and Suggestions for Authors
The manuscript describes a molecular biology investigation of weaning stress in piglets. The manuscript is generally well-written and easy to understand. There are a few considerations for improvement of the manuscript:
The results should be presented without interpretation. Thus, the entire section should be rewritten and discussion moved to the appropriate section.
Line 363 - Provide details about housing conditions and housing groups. What were the weaned pigs fed?
Author Response
Response to Reviewer 1 Comments
Comment: The manuscript describes a molecular biology investigation of weaning stress in piglets. The manuscript is generally well-written and easy to understand. There are a few considerations for improvement of the manuscript.
Response: Thank you very much for your valuable suggestion. We have seriously revised this manuscript and tried our best to address all the concerns. Below is our point-by-point response to your comments. In addition, there are some minor modifications don’t list in the response letter, but they have been shown in highlight in the revised manuscript.
Point 1: The results should be presented without interpretation. Thus, the entire section should be rewritten and discussion moved to the appropriate section.
Response 1: Thank you very much for your valuable suggestion. In response to your comment, we have revised the results section to ensure that it only presents the findings objectively, without interpretation. Any interpretative content has been relocated to the discussion section for appropriate analysis and elaboration.
Since a lower diarrhoea index is considered to be superior, these results suggest that Min pigs are less susceptible to weaning diarrhoea than Landrace pigs are.(Line 70)
As shown in Figure 1, after weaning, the intestinal mucosal structure of piglets changes, characterised by villus atrophy and rupture. , which can lead to impaired nutrient absorp-tion. Moreover, increased intestinal permeability makes it easier for exogenous factors to invade, thereby exacerbating intestinal inflammation. (Line 75-76)
In summary, weaning stress can affect the integrity of the intestinal mucosa in piglets, and the severity of intestinal damage is an important factor contributing to diarrhoea. Ad-ditionally, Min pigs have a lower stress response and less intestinal damage than Land-race pigs do, indicating that Min pigs possess greater intestinal resistance to damage. (Line 86)
In conclusion, the number of goblet cells is influenced by weaning stress, but Min pigs have a greater ability to respond to stress, as indicated by their better mucus barrier function.(Line 115)
In conclusion, weaning stress also affects goblet cell volumes. The goblet cell volumes in Min pigs are generally greater than those in Landrace pigs, suggesting that Min pigs may have better mucin secretion ability after weaning, which helps maintain better intestinal barrier function than that of Landrace pigs.(Line 133)
Point 2: Line 363 - Provide details about housing conditions and housing groups. What were the weaned pigs fed?
Response 2: Thank you very much for your valuable suggestion. According to the suggestions, we have added detailed information regarding the housing conditions, grouping, and diet of the weaned piglets. Specifically, we provided descriptions of the farrowing pen environment, rearing conditions after weaning, and the nutritional composition of the diet fed to the piglets. These additions aim to clarify the experimental setup and ensure the reproducibility of our study.
4.1. Animal Feeding and Husbandry
Before rearing, the farrowing pens were thoroughly cleaned and disinfected. The piglets were housed with the sow in the farrowing pens until weaning at 35 days of age. To minimise environmental stress, the piglets were observed until 38 days of age and con-tinued to be reared in the farrowing pens after weaning. The stocking density was main-tained at approximately 0.5 m² per piglet. The pens were equipped with heating lamps to ensure a temperature of 28–30°C, and the relative humidity was controlled at 60–70%. The piglets were fed twice daily with a standard commercial diet, and clean drinking water was provided ad libitum. The dietary nutrient levels were as follows: crude protein, 17.5%; net energy, 1.4 MJ/kg; lysine, 1.4%; total calcium, 0.6%; and phytate phosphate, 0.35%. Feed and water cleanliness were ensured throughout the experimental period.
A total of 108 piglets (48 Min piglets and 60 Landrace piglets) were randomly selected for the study. The farrowing pens and feeding equipment were cleaned and disinfected regularly to prevent disease. No antibiotics or probiotics were administered to avoid influencing the gut microbiota. (marked in red in Line 295-308)

Reviewer 2 Report
Comments and Suggestions for Authors
To the Authors
Weaning-induced diarrhoea (WID), with more than 50% incidence and apx 40% of mortality, is the primary stress-related disease that compromises the survival and growth of piglets. On the other hand, Mucin 2 (MUC2), a large and highly glycosylated protein, is key to i) binding sIgA and antimicrobial peptides; ii) providing adhesion sites for beneficial gut bacteria; iii) intestinal transport and absorption of nutrients. The Authors examine the susceptibility to WID and the intestinal mucus barrier and immune function in Min pigs, a high quality China breed, vs Landrace pigs. The results indicate that, as compared to the Landrace breed, the Min breed show a lower susceptibility to WID, a lower intestinal injury score, a higher number and a larger size of goblet cells, a higher duodenal and jejunal MUC2 mRNA expression in weaned animals. All these features would converge in a higher protection against the WID. The Authors focus their attention to the high MUC2 expression by speculating that it may serve as an important physiological basis for the lower intestinal injury scores and milder diarrhoea symptoms observed in Min pigs. While the number and quality of the experimental work made by the Authors are to be appreciated, a number of issues should be addressed prior to re-considering the present study for publication.
Major points
1. WID is just one model of stress, and Min breed is just one breed. The Authors are encouraged to discuss more deeply how the results of their beautiful experimental work on this specific model could be generalized in order to improve the current understanding of the response to stress at the molecular level. In particular, given the selected title, the Authors are encouraged to focus the discussion on the molecular role of MUC2.
2. Microbiota and gut-brain-axis could also play a role in the intestinal response to stress of WID. Have the Authors explored this point? Please clarify and/or justify.
Minor Points
1. It is unclear whether the title would accurately reflect the study content. Please clarify and/or justify.
2. The Abstract should be more focused.
3. Introduction is quite lengthy and confusing. Often the Authors are following the style of an academic lesson or a student thesis instead of an introduction to an experimental work. Hence, I would suggest rephrasing this section, shortening it up and focusing more clearly on the study objectives.
4. Discussion should be shortened and more focused on the interpretation and relevance of the results.
5. Figures should be self-explanatory. The Authors are encouraged to provide for each figure a short legend briefly explaining the displayed results instead of simply describing the elements of the graphics.
6. References list should be updated. If I am not mistaken, only 6 out of 48 cited Refs, i.e., 12.5% have been published in the last 5 years. Please clarify and/or justify.
7. Although the language does not prevent the understanding of the study content, a revision of the style could significantly improve the readability and impact of the study.
Author Response
Response to Reviewer 2 Comments
Comment: Weaning-induced diarrhoea (WID), with more than 50% incidence and apx 40% of mortality, is the primary stress-related disease that compromises the survival and growth of piglets. On the other hand, Mucin 2 (MUC2), a large and highly glycosylated protein, is key to i) binding sIgA and antimicrobial peptides; ii) providing adhesion sites for beneficial gut bacteria; iii) intestinal transport and absorption of nutrients. The Authors examine the susceptibility to WID and the intestinal mucus barrier and immune function in Min pigs, a high quality China breed, vs Landrace pigs. The results indicate that, as compared to the Landrace breed, the Min breed show a lower susceptibility to WID, a lower intestinal injury score, a higher number and a larger size of goblet cells, a higher duodenal and jejunal MUC2 mRNA expression in weaned animals. All these features would converge in a higher protection against the WID. The Authors focus their attention to the high MUC2 expression by speculating that it may serve as an important physiological basis for the lower intestinal injury scores and milder diarrhoea symptoms observed in Min pigs. While the number and quality of the experimental work made by the Authors are to be appreciated, a number of issues should be addressed prior to re-considering the present study for publication.
Response: Thank you very much for your valuable suggestion. Thank you very much for your valuable and insightful comments. Your feedback has greatly helped us to identify areas for improvement and enhance the quality of our manuscript. We have carefully revised the manuscript to address all the concerns raised and provided detailed responses to each of your comments below. Additionally, minor modifications not listed in this response letter have been highlighted in the revised manuscript for your reference.
Point 1: WID is just one model of stress, and Min breed is just one breed. The Authors are encouraged to discuss more deeply how the results of their beautiful experimental work on this specific model could be generalized in order to improve the current understanding of the response to stress at the molecular level. In particular, given the selected title, the Authors are encouraged to focus the discussion on the molecular role of MUC2.
Response 1: Thank you for your insightful comments regarding the generalization of our findings and the focus on MUC2’s molecular role. We appreciate your suggestion to deepen the discussion. In response, we have revised the discussion section to better emphasize the broader implications of our results beyond the specific model of weaning stress in Min pigs. We have included a more detailed analysis of the molecular mechanisms of MUC2 and its relevance to stress responses, which we hope enhances the overall impact of our study. We believe these revisions address your concerns effectively.
Point 2: Microbiota and gut-brain-axis could also play a role in the intestinal response to stress of WID. Have the Authors explored this point? Please clarify and/or justify.
Response 2: Thank you for your insightful comment regarding the potential role of microbiota and the gut-brain axis in the intestinal response to weaning-induced diarrhea (WID). We agree that these factors are critical in the broader context of gut health and stress responses. In the present study, we focused on the role of MUC2 and the intestinal mucus barrier in mitigating the effects of weaning stress, particularly in the context of differences between Min and Landrace pigs. Our experimental design and data collection were specifically tailored to investigate intestinal mucosal integrity and goblet cell activity, without incorporating analyses of microbiota or gut-brain interactions. We acknowledge that exploring the interplay between microbiota, the gut-brain axis, and intestinal barrier function could provide valuable insights into the mechanisms underlying WID. Future research could expand on our findings by integrating microbiota profiling and gut-brain axis assessments to better understand their contributions to stress responses and diarrhoea incidence.
Point 3: It is unclear whether the title would accurately reflect the study content. Please clarify and/or justify.
Response 3: Thank you for your insightful feedback regarding the title of our study. We chose "Exploring the Link between Mucin 2 and Weaning Stress-Related Diarrhoea in Piglets" to emphasize the central focus on MUC2 and its relationship with weaning stress, particularly in the context of diarrhoea in piglets. We believe this title accurately reflects the study's objectives and findings. However, we are open to suggestions for improvement if you have specific recommendations. We appreciate your guidance on this matter.
Point 4: The Abstract should be more focused.
Response 4: Thank you very much for your valuable suggestion. According to the suggestions, we have revised the abstract to enhance its focus on the key findings and implications of our study regarding MUC2 and weaning-induced diarrhoea in piglets.
Intestinal tissues were collected, and goblet cell numbers, sizes, and degrees of intestinal injury were observed and recorded. Intestinal tissue MUC2 mRNA and protein expression were analysed via quantitative real-time PCR (qRT‒PCR) and Western blotting. Min pigs presented significantly lower diarrhoea rates and intestinal injury scores than Landrace pigs (p < 0.01). The intestinal injury scores in the weaned diarrhoea group were significantly greater than those in the unweaned groups (p < 0.05), with Min pigs consistently exhibiting lower injury scores than Landrace pigs. Specifically, unweaned Min pigs presented significantly greater duodenal MUC2 mRNA (p < 0.05), and weaned healthy Min pigs presented notably greater expression in both the duodenum and jejunum (p < 0.01). These findings reveal enhanced intestinal protection against weaning stress and diarrhoea in Min pigs, with elevated MUC2 levels likely contributing to lower injury scores and milder symptoms, thus highlighting the influence of genetic differences.(marked in red in Line 16-27)
Point 5: Introduction is quite lengthy and confusing. Often the Authors are following the style of an academic lesson or a student thesis instead of an introduction to an experimental work. Hence, I would suggest rephrasing this section, shortening it up and focusing more clearly on the study objectives.
Response 5: Thank you very much for your valuable suggestion. According to the suggestions, we have carefully revised the Introduction section to make it more concise and focused. Redundant background information has been removed, and the study objectives are now presented more clearly to align with the experimental focus of the work. We believe these revisions address your concerns and improve the overall readability of the section.
Weaning is an inevitable stage in growth but can disrupt internal homeostasis due to separation from sows and changes in nutrition and the environment [1,2]. This imbalance can lead to adverse symptoms, particularly weaning-induced diarrhoea [3], which is the primary stress-related disease affecting piglet survival and growth. (marked in red in Line 31-34)
More than 20 types of mucins have been identified, categorised as secreted or mem-brane-bound [7]. Secreted mucins, including MUC2, MUC5AC, MUC5B, MUC6, and MUC19, form a protective mucus gel on intestinal epithelial cells [8,9]. (marked in red in Line 43-46)
Once released into the intestinal lumen, MUC2 forms a polymer network, ensuring structural stability and serving as a binding site for secretory immunoglobulin A (sIgA) and antimicrobial peptides, enhancing immune barrier function [16,17]. MUC2 also sup-ports the transport and absorption of nutrients [18], highlighting its clinical importance in maintaining intestinal health. (marked in red in Line 55-59)
This study aimed to investigate whether Min pigs possess a more robust intestinal mu-cus barrier and immune function than other breeds do and to determine the role of these intestinal characteristics in their susceptibility to postweaning diarrhoea. (marked in red in Line 61-64)
Point 6: Discussion should be shortened and more focused on the interpretation and relevance of the results.
Response 6: Thank you very much for your valuable suggestion. According to the suggestions, we have carefully revised the Discussion section to make it more concise and focused. Redundant information has been removed, and greater emphasis has been placed on the interpretation and relevance of the results. We believe these revisions address your concerns and enhance the clarity and impact of this section.
Histological analysis revealed varying degrees of intestinal damage in both breeds after weaning, with the small intestine being the most affected. Damage included villous rup-ture, epithelial necrosis, and crypt hyperplasia, with diarrhoea groups exhibiting more severe damage. Min pigs consistently presented lower intestinal damage scores than Landrace pigs. The integrity of the intestinal mucosa is crucial for nutrient absorption and defence against harmful substances [21,22]. (marked in red in Line 210-216)
Weaning stress leads to a reduction in villus height and an increase in crypt depth [26]. Overall, weaning stress damages the intestinal mucosa in piglets, causing unfavourable changes in intestinal morphology [27]. This disruption impairs nutrient and water ab-sorption, contributing to diarrhoea in piglets [28].(marked in red in Line 218-221)
The mucus layer formed by mucins, primarily MUC2, is secreted by goblet cells and serves as the first line of intestinal defence by preventing pathogen invasion and main-taining mucosal integrity [29-31]. Damage to goblet cells reduces mucin secretion, weak-ening the mucus barrier and increasing susceptibility to inflammation [32,33]. Studies have linked decreased mucin synthesis to goblet cell depletion in conditions such as in-flammatory bowel disease and necrotising enterocolitis [34-36]. Goblet cells also play an essential role in neonatal intestinal immunity before passive immunity is established [37]. (marked in red in Line 222-228)
In the healthy Min pig group, the number of goblet cells was greater than that in the diarrhoea group, whereas in the Landrace piglets, the number of goblet cells was significantly lower, increasing the risk of intestinal damage. (marked in red in Line 230-233)
Compared with Landrace pigs, healthy weaned Min pigs presented greater numbers and volumes of goblet cells, especially in the duodenum and jejunum. (marked in red in Line 237-239)
MUC2 not only is vital for maintaining mucosal integrity in the context of wean-ing-induced diarrhoea but also has been associated with other stress-related conditions, such as necrotising enterocolitis and heat stress [52-56]. In these cases, MUC2 plays a cru-cial role in maintaining mucosal integrity and preventing systemic inflammation [16]. These findings highlight MUC2 as a universal indicator of intestinal barrier function [57], with potential applications for enhancing gut health not only in various livestock species but also in humans [58].(marked in red in Line 252-258)
Jejunal patterns of expression varied between Min pigs and Landrace pigs. Compared with those in the unweaned and weaned healthy groups, the MUC2 mRNA and protein levels in the Min pig diarrhoea group were lower, whereas those in the Landrace diar-rhoea group were greater than those in both healthy groups. (marked in red in Line 260-264)
Although the Landrace diarrhoea group presented greater goblet cell numbers and vol-umes than the Min pig group did, acute MUC2 secretion did not significantly improve their defence ability, as Min pigs presented lower diarrhoea rates and indices. Healthy Min pigs presented greater goblet cell numbers, volumes, and MUC2 expression, whereas diarrhoeal Min pigs presented lower levels than Landrace pigs did. (marked in red in Line 267-272)
Additionally, integrating MUC2 regulation into nutritional strategies such as the inclu-sion of prebiotics or amino acids to support mucus production could help mitigate stress-induced intestinal damage [59]. Extending the experimental period could provide further insights into the recovery process of intestinal barrier function and its timing, of-fering a theoretical foundation for the development of antistress strategies and improving intestinal health in pigs. Investigating the role of MUC2 in these complex stress models will reveal its broader applicability in promoting livestock health and productivity. (marked in red in Line 283-289)
Point 7: Figures should be self-explanatory. The Authors are encouraged to provide for each figure a short legend briefly explaining the displayed results instead of simply describing the elements of the graphics.
Response 7: Thank you very much for your valuable suggestion. According to the suggestions, we have revised the figure legends to ensure they are more self-explanatory. Each legend now includes a brief explanation of the displayed results, in addition to describing the elements of the graphics. We believe this improvement makes the figures more informative and accessible to readers.
Figure 1. Intestinal structures of Landrace and Min pigs. In the weaning diarrhoea group, Landrace pigs presented significant villi rupture (black arrows) and mucosal shedding (red arrow). Healthy weaned Landrace pigs exhibit mucosal shedding (red arrow), whereas Min pigs exhibit less damage and villi atrophy (yellow arrow). (marked in red in Line 88-91)
Figure 2. Injury scores of the duodenum, jejunum, ileum, and colons in Landrace and Min pigs. After weaning, both Min and Landrace piglets presented increased intestinal damage scores, with the diarrhoea groups showing significantly higher scores than the unweaned group, * p < 0.05, ** p < 0.01. (marked in red in Line 93-96)
Figure 3. Intestinal goblet cells of Landrace and Min pigs. PAS staining revealed numerous well-defined goblet cells in the intestine, with their cytoplasm filled with red-stained mucus granules. (marked in red in Line 135-136)
Figure 4. Numbers of goblet cells in the intestines of Landrace and Min pigs. After weaning, Landrace pigs presented a decrease in the number of goblet cells, whereas Min pigs presented higher goblet cell counts, indicating better intestinal health, * p < 0.05, ** p < 0.01, *** p < 0.001. (marked in red in Line 138-140)
Figure 5. Volumes of goblet cells in the intestines of Landrace and Min pigs. Volumes of goblet cells in the intestines of Landrace and Min pigs. After weaning, the goblet cell volume increased in healthy pigs but decreased in the diarrhoea group. Min pigs presented greater goblet cell volumes than Landrace pigs did, especially in the duodenum and jejunum, * p < 0.05, ** p < 0.01, *** p < 0.001. (marked in red in Line 142-145)
Figure 6. Intestinal MUC2 mRNA expression pattern. Min pigs presented increased MUC2 expression in the jejunum after weaning, whereas MUC2 expression decreased in the diarrhoea group. Min pigs presented greater overall MUC2 expression than Landrace pigs did in all segments, * p < 0.05, ** p < 0.01, *** p < 0.001. (marked in red in Line 163-166)
Figure 7. Protein bands and relative MUC2 expression levels in the duodena, jejuna, ilea, and colons. (A): Protein bands; (B): Relative expression levels. Min pigs presented increased duodenal MUC2 ex-pression after weaning, whereas Landrace pigs presented decreased MUC2 expression. Overall, Min pigs presented greater MUC2 expression than Landrace pigs did, especially in the duodenum. * p < 0.05, ** p < 0.01. (marked in red in Line 188-191)
Figure 8. Intestinal MUC2 protein expression and localisation in the nonweaned group. MUC2 was ex-pressed primarily in the epithelial layer of the villi and lamina propria of the small intestine. No significant localisation changes occurred due to weaning stress. (marked in red in Line 203-205)
Point 8: References list should be updated. If I am not mistaken, only 6 out of 48 cited Refs, i.e., 12.5% have been published in the last 5 years. Please clarify and/or justify.
Response 8: Thank you very much for your valuable suggestion. According to the suggestions, we have carefully reviewed and updated the reference list. It is true that a portion of the cited references are older, primarily because certain foundational theories and methods in our research have been well-established over time, and these classic works continue to hold significant relevance. However, we acknowledge the importance of more recent studies and will include additional references published within the last five years to provide a more comprehensive overview of the latest advancements in the field.
Point 9: Although the language does not prevent the understanding of the study content, a revision of the style could significantly improve the readability and impact of the study.
Response 9: Thank you for your insightful feedback regarding the language and style of the manuscript. According to the suggestions, we have made revisions to improve clarity and readability. We hope these changes will enhance the overall presentation of the study while maintaining the integrity of the content. Your input is invaluable in helping us improve our work.

Round 2
Reviewer 1 Report
Comments and Suggestions for Authors
This manuscript describing a molecular biology investigation of weaning stress in piglets is well-written, providing the expected and necessary information.
Reviewer 2 Report
Comments and Suggestions for Authors
To the Authors
The Authors of the Ms. ID ijms-3386832 entitled “Exploring the Link between Mucin 2 and Weaning Stress-Related Diarrhoea in Piglets” have carefully and satisfactorily addressed all my prior points of criticism. I feel that the revised version of the Ms is significantly improved in terms of scientific communication and readability.